# Backscatter Assisted NOMA-PLNC Based Wireless Networks

**DOI:** 10.3390/s21227589

**Published:** 2021-11-15

**Authors:** Samikkannu Rajkumar, Dushantha Nalin K. Jayakody

**Affiliations:** 1Centre for Telecommunication Research, School of Engineering, Sri Lanka Technological Campus, Padukka 10500, Sri Lanka; rajkumars@sltc.ac.lk; 2School of Computer Science and Robotics, National Research Tomsk Polytechnic University, 634050 Tomsk, Russia

**Keywords:** non-orthogonal multiple access, physical layer network coding, ambient backscattering

## Abstract

In this paper, sum capacity maximization of the non-orthogonal multiple access (NOMA)-based wireless network is studied in the presence of ambient backscattering (ABS). Assuming that ABS is located next to far nodes, it improves the signal strength of far node cluster. By applying suitable successive interference cancellation (SIC) operation, far node cluster act as an internet of things (IoT) reader. Moreover, to improve the uplink performance of the nodes, a physical layer network coding (PLNC) scheme is applied in the proposed network. Power optimization is employed at the access point (AP) to enhance the downlink performance with total transmit power constraint and minimum data rate requirement per user constraint using Lagrangian’s function. In addition, end-to-end outage performance of the proposed wireless network is analyzed to enhance each wireless link capacity. Numerical results evident that the outage performance of the proposed network is significantly improved while using the ABS. Furthermore, the average bit error rate (BER) performance of the proposed wireless network is studied to improve the reliability. Simulation results are presented to validate the analytical expressions.

## 1. Introduction

In recent years, the usage of smartphones with internet supports increased rapidly due to its variety of real-time applications such as online education, e-healthcare and e-business etc. which causes an exponential growth on mobile data traffic in the wireless environment [1]. Therefore, it is necessary to develop a network model to maintain the mobile data traffic in fifth generation and beyond (5GB) and to support massive connectivity for the low power internet of things (IoT) devices.

Non-orthogonal multiple access (NOMA) scheme is identified as one of the most prominent technology in 5G wireless network. Main features of NOMA include its capability to support large number of mobile connections and high spectral efficiency [2]. In particular, the power domain NOMA serves for several users with same time and frequency resources provided with different power levels according to the channel conditions [3,4]. Hence, the proper power allocation among the mobile users is another major problem in NOMA-based wireless network. Basic operation of the power domain NOMA includes the superposition coding (SC) to combine the transmit signals and the successive interference cancellation (SIC) to separate the signals at the receiver.

Since the NOMA users are sharing the common channel, the user with poor channel condition may not decode its data properly. To address this issue, many research works has been proposed through allocating power depend on the channel conditions. Joint power allocation and time switching operation is proposed in simultaneous wireless information and power transfer (SWIPT)-based NOMA system [5]. In that model, power optimization is applied to maximize the energy efficiency of NOMA system while satisfying the constraints such as maximum transmit power, minimum data rate requirement per user and minimum harvested energy per user. In [6], optimal power allocation in the downlink NOMA system is analyzed to maximize the sum-rate and to maximize the energy efficiency with QoS constraints. Similarly, optimal resource allocation in the downlink multi-user NOMA system is analyzed in [7] to maximize the sum-rate. Using projected gradient descent algorithm and dynamic programming, sum-rate maximization of the multi-carrier NOMA is analyzed in [8]. By combining cognitive radio (CR) and NOMA, a network model is proposed to improve spectrum efficiency in 5G communications [9,10]. Cellular-based vehicular networks is proposed using CR and NOMA scheme to support the massive connections, high reliability and spectral efficiency in [11].

Physical layer network coding (PLNC) is another popular scheme in wireless communication, and it works based on the broadcast nature of the wireless medium [12,13]. In this scheme, the network coding concept is employed, and it allows the two users to exchange their information within the two time slots rather than three time slots in the conventional network coding scheme. Sum-rate performance of the PLNC-based two-way relay network is analyzed in [14]. Outage performance of the PLNC-based large scale cellular network is analyzed in the presence of intercell interference (ICI) in [15]. Similarly, outage performance of the PLNC-based full-duplex cognitive radio network is analyzed under Nakagami-*m* fading environment in [16]. Recently, a few works have been proposed combining PLNC and NOMA schemes. For instance, outage performance of the full-duplex cooperative NOMA system with PLNC is analyzed in [17]. Outage performance of the two-way relay-based NOMA system is analyzed with perfect and imperfect-SIC through independent non-identically distributed (ind) fading channels in [18]. In this paper, a new low power wireless network model is proposed by combining PLNC and NOMA schemes. Comparison of NOMA and NOMA with PLNC is given in Table 1.

### 1.1. Motivation

Although the optimal power allocation in NOMA-based wireless network will improve the sum capacity, the quality of the far user is still not healthy due to the poor signal strength. By considering this issue, the concept of ambient backscattering (ABS) [19] is applied in this paper which help to improve the signal strength of the far user. ABS is the technique which generally used in the low power device communication, and it operates without any specific RF transmitters however, it uses the existing ambient radio frequency (RF) signals for transmission. For instance, ambient backscatter transmitter (example, tag) can communicate with backscatter receiver (example, reader) using surrounding ambient RF sources examples TV towers, FM towers, cellular base stations and Wireless Fidelity (Wi-Fi) access points etc. [19]. Outage probability of an ambient backscatter system with pair of tag and reader is analyzed in [20]. Backscatter communication network model is proposed using WiFi, hence backscatter tag can be read by WiFi devices [21]. A wireless protocol is proposed using large scale backscatter devices and addressed timing, frequency synchronization and near-far problem [22]. A versatile MIMO backscatter is proposed to leverage the diversity features of MIMO, therefore it dramatically decreases bit error rate (BER) in [23]. A OFDM backscatter system is proposed, and it uses a sample-level modulation to avoid the phase offset created by a tag and hence improves the throughput in [24]. Similarly, LTE backscatter system is developed to leverage the continuous LTE ambient traffic for ubiquitous, high throughput and low latency in [25]. Recently, a few works have been proposed incorporating NOMA and backscattering. In [26], closed form expressions for the outage probability of the backscatter-NOMA is presented and studied how the cellular systems and IoT networks are working together. Outage performance, expected rates, and diversity-multiplexing tradeoff (DMT) of the backscatter cooperation NOMA is studied in [27]. Hybrid time division multiple access (TDMA)-based power domain NOMA is proposed in IoT sensors network in the presence of Backscatterer [28]. Similarly, the ergodic capacity and outage performance of the bidirectional NOMA-SWIPT enabled IoT wireless network is analyzed in [29]. In this paper, a novel 5G wireless network model is proposed using NOMA and PLNC schemes with ABS. It is assumed that the PLNC scheme is used in the uplink and the NOMA scheme used in the downlink.

### 1.2. Contributions

The main contributions of this paper are:proposal of low power 5G wireless network using NOMA and PLNC schemes with ambient backscattering.derivation for optimal power allocation to the near node and the far node with ABS.closed form expression for the end-to-end outage probability of the proposed wireless network with ABS.closed form expression for the end-to-end average BER of the proposed wireless network with ABS.

The rest of the paper is organized as follows: In Section 2, the system model of the proposed wireless network with ABS is discussed. The derivation for the optimal power allocation of the downlink NOMA with ABS is presented in Section 3. The end-to-end outage probability expression is derived in Section 4 and average BER expression is given in Section 5. Numerical results of the proposed wireless network is presented in Section 6 and the concluding remarks are given in Section 7.

## 2. System Model

Consider a proposed wireless network which consists of an access point (AP), one near node pair and one far node pair as shown in Figure 1. It is assumed that an ABS transmitter exists in this model near to the far node pair which will help to improve the signal strength of the far node pair. The network model is designed using the PLNC scheme such that nodes can exchange their information with three time slots. The detailed time slot management of the proposed network is shown in Figure 2a. Using orthogonal multiple access channel (OMAC), the near node pair and far node pair send their data to the AP in first time slot and second time slot, respectively. At the AP, nodes data will be detected, decoded and then XORed using the PLNC mapping. Then, the resulting PLNC encoded signals are combined with different power levels using superposition coding as shown in Figure 2b. Then, the resulting NOMA signal is transmitted back to the same node pairs simultaneously over non-orthogonal broadcast channel (NOBC) in the third time slot. In this model, far node pair acting as an IoT reader using suitable SIC operation as shown in Figure 2c. Far node pair first decode the PLNC signal then performing the PLNC demapping to obtain the original transmitted signal. Then using SIC, it removes its own message then decode backscatter message.

The data transmission between the nodes in this wireless network are performed over the two phases such as OMAC phase and NOBC phase.

### 2.1. OMAC Phase

In first time slot, near node pair send the data to the AP. The time domain received signal $z_{r,1}\left(t\right)$ over OMAC link at the AP (i.e., relay) is expressed as
(1)$$z_{r,1}\left(t\right)\;=\;\sqrt{P_{t}}x_{1}\left(t\right)h_{1}\left(t\right)\;+\;\sqrt{P_{t}}x_{2}\left(t\right)h_{2}\left(t\right)+n_{1}\left(t\right),$$
where $P_{t}$ is the transmit power of the node $n_{1},n_{2},n_{3}$ and $n_{4}$, $x_{1}\left(t\right)$ and $x_{2}\left(t\right)$ are the binary phase shift keying (BPSK) modulated signals send by the node 1 and node 2, respectively. The average values of the transmit signals are $E({\left|x_{1}\left(t\right)\right|}^{2})$ = $E({\left|x_{2}\left(t\right)\right|}^{2})$ = 1 and $h_{1}\left(t\right)$, $h_{2}\left(t\right)$ are the channel coefficients between node 1 to AP and node 2 to AP whose magnitudes follows the exponential distribution with mean zero and variances $\eta_{1}$ and $\eta_{2}$ respectively [18], $n_{1}$ is the independent identically distributed (iid) complex additive white Gaussian noise (AWGN) with mean μ and the variance $\sigma^{2}$, denoted by $n_{1}\left(t\right)∼\mathrm{CN}\left(\mu ,\;\sigma^{2}\right)$. Similarly, the far node pair send the data to the AP in the second time slot. The time domain received signal $z_{r,2}\left(t\right)$ from the far node pair is expressed as
(2)$$z_{r,2}\left(t\right)\;=\;\sqrt{P_{t}}x_{3}\left(t\right)h_{3}\left(t\right)+\sqrt{P_{t}}x_{4}\left(t\right)h_{4}\left(t\right)+n_{2}\left(t\right),$$
where $x_{3}\left(t\right)$ and $x_{4}\left(t\right)$ are the BPSK signals transmitted from the node 3 and node 4 of the far node pair, $h_{3}\left(t\right)$, $h_{4}\left(t\right)$ are the channel coefficients between node 3 to AP and node 4 to AP whose magnitudes follows the exponential distribution with mean zero and variances $\eta_{3}$ and $\eta_{4}$ respectively, $n_{2}\left(t\right)$ is the iid Gaussian noise.

### 2.2. NOBC Phase

In this phase, the received signals from the near node pair and far node pair are detected, decoded to obtain the PLNC signals using maximum likelihood (ML) detection. Thus, by simply multiplying near node detected signals and far node detected signals, the PLNC signals can be computed as [30]
(3)$${\hat{x}}_{r_{1}}\left(t\right)\;=\;{\hat{x}}_{1}\left(t\right)\times {\hat{x}}_{2}\left(t\right);\;{\hat{x}}_{r_{2}}\left(t\right)\;=\;{\hat{x}}_{3}\left(t\right)\times {\hat{x}}_{4}\left(t\right),$$
where ${\hat{x}}_{m}\left(t\right),\;m\in \left(1,2,3,4\right)$ are detected signals at relay. Then, these signals are added with different power levels $\alpha_{n}$ where *n*∈(1,2) using NOMA scheme and it can be written as
(4)$${\hat{x}}_{r_{3}}\left(t\right)\;=\;\alpha_{1}\sqrt{P_{r}}{\hat{x}}_{r_{1}}\left(t\right)\;+\;\alpha_{2}\sqrt{P_{r}}{\hat{x}}_{r_{2}}\left(t\right),$$
where $P_{r}$ is the AP transmit power, which is chosen such that it will be equal to the node transmit power. The power allocation coefficient $\alpha_{n},\;n\in \left(1,2\right)$ satisfy the condition $\alpha_{1}+\alpha_{2}$ = 1 and for convenient we used the notations $\alpha_{1}\sqrt{P_{r}}\;=\;p_{1}$, and $\alpha_{2}\sqrt{P_{r}}\;=\;p_{2}$ in rest of the paper.

In downlink, the AP transmits ${\hat{x}}_{r3}\left(t\right)$ to all the four nodes simultaneously over NOBC link. The time domain signal $z_{n_{1}}\left(t\right)$ and $z_{n_{2}}\left(t\right)$ received by the near node pair can be written as
(5)$$z_{n_{1}}\left(t\right)\;=\;{\hat{x}}_{r_{3}}\left(t\right)g_{1}\left(t\right)+n_{n_{1}}\left(t\right),$$
(6)$$z_{n_{2}}\left(t\right)\;=\;{\hat{x}}_{r_{3}}\left(t\right)g_{2}\left(t\right)+n_{n_{2}}\left(t\right),$$
where $g_{1}\left(t\right)$ and $g_{2}\left(t\right)$ are the channel coefficients between the AP to the near node 1 and the AP to the near node 2 and modeled as an ind random variable which follows the exponential distribution with mean zero and variances $\eta_{n_{1}}=d_{1}^{-\alpha }$ and $\eta_{n_{2}}=d_{2}^{-\alpha }$ respectively, where $d_{1},d_{2}$ are distances between near nodes and AP [18], $n_{n_{1}}\left(t\right)$ and $n_{n_{2}}\left(t\right)$ are the iid Gaussian noise. Similarly, the received signals $z_{f_{1}}\left(t\right)$ and $z_{f_{2}}\left(t\right)$ by the far node pair are written as
(7)$$z_{f_{1}}\left(t\right)\;=\;{\hat{x}}_{r_{3}}\left(t\right)g_{3}\left(t\right)+\beta e_{3}\left(t\right)b_{3}\left(t\right){\hat{x}}_{r_{3}}\left(t\right)x_{b}\left(t\right)+n_{f_{1}}\left(t\right),$$
(8)$$z_{f_{2}}\left(t\right)\;=\;{\hat{x}}_{r_{3}}\left(t\right)g_{4}\left(t\right)+\beta e_{4}\left(t\right)b_{4}\left(t\right){\hat{x}}_{r_{3}}\left(t\right)x_{b}\left(t\right)+n_{f_{2}}\left(t\right),$$
where β is the reflection coefficient used to normalize $x_{b}\left(t\right)$ [26] which is backscatter own message and its average value is $E({\left|x_{b}\left(t\right)\right|}^{2})$ = 1, $g_{3}\left(t\right)$ and $g_{4}\left(t\right)$ are the channel coefficients between the AP to the far node 1 and the AP to the far node 2 with mean zero and variances $\eta_{f_{1}}=d_{3}^{-\alpha }$ and $\eta_{f_{2}}=d_{4}^{-\alpha }$ respectively, where $d_{3},d_{4}$ are distances between far nodes and AP, $e_{3}\left(t\right)$, $b_{3}\left(t\right)$, $e_{4}\left(t\right)$ and $b_{4}\left(t\right)$ are ambient backscattering channel coefficients, $n_{f_{1}}$ and $n_{f_{2}}\left(t\right)$ are the Gaussian noise.

## 3. Power Optimization in Downlink NOMA

Using optimal power allocation, the sum capacity of the proposed network is enhanced at the downlink. The objective function is formed by maximizing the sum capacity while maintaining the constraint for the total transmit power of the AP and the minimum data rate requirement for each node. Mathematically, the problem can be formulated as follows
(9)$$\begin{array}{c} \underset{p_{1},p_{2}}{\max_{︸}}\;\sum_{m=1}^{4}\log\left(1+\gamma_{m}^{d}\right)\\ s.t.\;\sum_{m=1}^{4}\log\left(1+\gamma_{m}^{d}\right)-C_{min}\;\geq \;0\\ p_{1}+p_{2}\;\leq \;p_{max}, \end{array}$$
where $p_{max}$ is the maximum transmit power of the AP, $C_{min}$ is the minimum required data rate for each node, $\gamma_{1}^{d}$ and $\gamma_{2}^{d}$ are the downlink signal to noise ratio (SNR) of the near node pair, $\gamma_{3}^{d}$ and $\gamma_{4}^{d}$ are the downlink signal to interference plus noise ratio (SINR) of the far node pair. The SINR of the far node to decode the near node can be written as
(10)$$\gamma_{m,n\to f}^{d}=\frac{{\left|g_{m}\right|}^{2}p_{2}}{{\left|g_{m}\right|}^{2}p_{1}+\sigma^{2}};\;m\in \left(1,2\right),$$

After removing the far node signal, the SNR of the near node can be computed.
(11)$$\gamma_{m}^{d}=\frac{p_{1}{\left|g_{m}\right|}^{2}}{\sigma^{2}};\;m\in \left(1,2\right),$$

Similarly, SINR expression for the far node pair can be expressed as
(12)$$\gamma_{m}^{d}=\frac{{\left|g_{m}\right|}^{2}p_{2}+\beta {\left|e_{m}\right|}^{2}{\left|b_{m}\right|}^{2}p_{2}}{{\left|g_{m}\right|}^{2}p_{1}+\beta {\left|e_{m}\right|}^{2}{\left|b_{m}\right|}^{2}p_{1}+\sigma^{2}};\;m\in \left(3,4\right),$$

It is noted that the far node achieves the diversity from the ABS signal. Furthermore, it can be designed such that it can decode the ABS signal after removing its own message. Thus, far node can also be used as an IoT reader [26]. The SNR of the far node to decode the ABS signal is given as
(13)$$\gamma_{m,x|x_{b}}^{d}=\frac{\beta {\left|e_{m}\right|}^{2}{\left|b_{m}\right|}^{2}p_{2}}{\sigma^{2}};\;m\in \left(3,4\right),$$

Since the constraints are linear and the objective function is concave [5], the equivalent problem becomes the convex maximization, it can be solved using Karush Kuhn Tucker (KKT) conditions [31]. Using Lagrangian multiplier’s, (9) can be rewritten as
(14)$$\begin{array}{c} L\left(p,\lambda ,\mu \right)=\sum_{m=1}^{4}\log\left(1+\gamma_{m}^{d}\right)\\ -\sum_{m=1}^{4}\lambda_{m}\left(C_{min}-\log\left(1+\gamma_{m}^{d}\right)\right)\\ -\mu \left(\left(p_{1}+p_{2}\right)-p_{max}\right), \end{array}$$
where $\lambda_{i}\geq 0,i\in \left(1,2,3,4\right)$ are the Lagrangian’s multipliers for the minimum rate constraints of near node and far node respectively and μ≥0 is the Lagrangian’s multiplier of the maximum transmit power. The problem can be modeled as dual objective function such that it maximizes the sum capacity while minimize the constraints. Mathematically, the first objective function to maximize the sum capacity can be written as
(15)$$a\left(\lambda ,\mu \right)=\underset{p_{1},p_{2}}{\max_{︸}}\;L\left(p,\lambda ,\mu \right),$$

Similarly, the dual objective function can be expressed as
(16)$$\begin{array}{c} \underset{\lambda_{1},\lambda_{2},\lambda_{3},\lambda_{4},\mu }{\min_{︸}}\;a\left(\lambda ,\mu \right), \end{array}$$
(17)$$\begin{array}{c} s.t.\;\sum_{m=1}^{4}\lambda_{m}\geq 0,\mu \geq 0, \end{array}$$

Using gradient descent algorithm, the optimal power $p_{1}$ and $p_{2}$ can be estimated. By substituting $\gamma_{m}^{d}$ in (14) and differentiating with respect to $p_{1}$ and $p_{2}$, the optimal powers $p_{1}$ and $p_{2}$ can be computed and either $p_{1}$ or $p_{2}$ is used for optimal power allocation.
(18)$$\begin{array}{c} \nabla L_{p_{1}}=\sum_{m=1}^{2}\frac{{\left|g_{m}\right|}^{2}}{\sigma^{2}+p_{1}{\left|g_{m}\right|}^{2}}\left(1+\lambda_{m}\right)\\ +\sum_{m=3}^{4}\frac{{\left|g_{m}\right|}^{2}+\beta {\left|e_{m}\right|}^{2}{\left|b_{m}\right|}^{2}}{{\left|g_{m}\right|}^{2}\left(p_{1}+p_{2}\right)+\beta {\left|e_{m}\right|}^{2}{\left|b_{m}\right|}^{2}\left(p_{1}+p_{2}\right)+\sigma^{2}}\times \\ \left(1+\lambda_{m}\right)-\sum_{m=3}^{4}\frac{{\left|g_{m}\right|}^{2}+\beta {\left|e_{m}\right|}^{2}{\left|b_{m}\right|}^{2}}{{\left|g_{m}\right|}^{2}p_{1}+\beta {\left|e_{m}\right|}^{2}{\left|b_{m}\right|}^{2}p_{1}+\sigma^{2}}\left(1+\lambda_{m}\right)-\mu , \end{array}$$
(19)$$\begin{array}{c} \nabla L_{p_{2}}=\sum_{m=3}^{4}\frac{{\left|g_{m}\right|}^{2}+\beta {\left|e_{m}\right|}^{2}{\left|b_{m}\right|}^{2}}{{\left|g_{3}\right|}^{2}\left(p_{1}+p_{2}\right)+\beta {\left|e_{m}\right|}^{2}{\left|b_{m}\right|}^{2}\left(p_{1}+p_{2}\right)+\sigma^{2}}\times \\ (1+\lambda_{m})-\mu , \end{array}$$

**Proof.** See Appendix A.    □

The power updating process can be performed in iterative manner. At *n*th iteration,
(20)$$p_{i}\left(n\right)=\left[p_{i}\left(n-1\right)+t\nabla L_{p_{i}}\right];\;i\in \left(1,2\right),$$
where *t* is the step size. In this way, the sub-gradient of the dual function can be computed as
(21)$$\nabla a_{\lambda_{m}}=\sum_{m=1}^{2}\log\left(1+\frac{p_{1}{\left|g_{m}\right|}^{2}}{\sigma^{2}}\right)-C_{min},$$
(22)$$\nabla a_{\lambda_{m}}=\sum_{m=3}^{4}\log\left(1+\frac{\left({\left|g_{m}\right|}^{2}p_{2}+\beta {\left|e_{m}\right|}^{2}{\left|b_{m}\right|}^{2}p_{2}x\right)}{\left({\left|g_{m}\right|}^{2}p_{1}+\beta {\left|e_{m}\right|}^{2}{\left|b_{m}\right|}^{2}p_{1}x+\sigma^{2}\right)}\right)-C_{min},$$
(23)$$\nabla a_{\mu }=p_{max}-\left(p_{1}+p_{2}\right),$$

Similarly, the Lagrangian multipliers are updated in iterative manner as given by
(24)$$\lambda_{m}\left(n\right)=\left[\lambda_{m}\left(n-1\right)+q_{m}\nabla a_{\lambda_{m}}\right]\;m\in \left(1,2,3,4\right),$$
(25)$$\mu \left(n\right)=[\mu \left(n-1\right)+l\nabla a_{\mu }].$$
where $q_{m},l$ denote the step sizes. Iterative algorithm to obtain the optimal power and the optimal Lagrangian’s multiplier’s is given in Algorithm 1.
  **Algorithm 1:** Iterative Optimal Power Allocation [5].   1. $\mathrm{initialize}\lambda_{1}\left(0\right)=0,\lambda_{2}\left(0\right)=0,\lambda_{3}\left(0\right)=0,\lambda_{4}\left(0\right)=0,\mu \left(0\right)=0,$   2. set the stopping criterion’s $\epsilon_{1}$ and $\epsilon_{2}$   3. Let n = 1, $\mathrm{initialize}\;p_{1}=\left(0\right)$ and $p_{2}=\left(0\right)$   4. updates $p_{1}$ and $p_{2}$ according to (20)   5. n = n + 1 repeat until convergence i.e., $∥\nabla L_{p}{∥}^{2}\leq \epsilon_{1}$   6. updates $\lambda_{1}\left(n\right),\lambda_{2}\left(n\right),\lambda_{3}\left(n\right),\lambda_{4}\left(n\right),\mu \left(n\right)$ according to (24) and (25)   7. Repeat until $\lambda_{1}\left(n\right),\lambda_{2}\left(n\right),\lambda_{3}\left(n\right),\lambda_{4}\left(n\right),\mu \left(n\right)$ converge.   8. $∥\lambda_{i}\left(n\right)-\lambda_{i}{\left(n-1\right)∥}^{2}\leq \epsilon_{2}\;i\in \left(1,2,3,4\right)$${\;∥\mu \left(n\right)-\mu \left(n-1\right)∥}^{2}\leq \epsilon_{2}$


## 4. Outage Probability Analysis

In this section, the outage performance of the proposed wireless network with ambient backscattering is analyzed. The outage performance of an OMAC link (i.e., uplink) per node and the NOBC link (i.e., downlink) per node is derived. Then, using an OMAC link outage, the end-to-end outage probability is derived.

### 4.1. Uplink Outage Probability: An OMAC Link

Let $\gamma_{1}^{u}\;$, $\gamma_{2}^{u}\;$, $\gamma_{3}^{u}\;$, $\gamma_{4}^{u}\;$ are the instantaneous uplink SNR of the received signals of node 1, node 2, node 3 and node 4 respectively at the AP and they can be written as
(26)$$\gamma_{1}^{u}\;=\;\frac{{\left|h_{1}\right|}^{2}\;P_{t}}{\sigma^{2}};\;\gamma_{2}^{u}\;=\;\frac{{\left|h_{2}\right|}^{2}\;P_{t}}{\sigma^{2}},$$
(27)$$\gamma_{3}^{u}\;=\;\frac{{\left|h_{3}\right|}^{2}\;P_{t}}{\sigma^{2}};\;\gamma_{4}^{u}\;=\;\frac{{\left|h_{4}\right|}^{2}\;P_{t}}{\sigma^{2}},$$

Using (26) and (27), the outage probability of an OMAC link can be expressed as
(28)$$Pr_{out}^{OMAC}\left(R\right)=Pr\left(min\left[min\left(\;\gamma_{1}^{u}\;,\gamma_{2}^{u}\;\right)\;,min\left(\;\gamma_{3}^{u}\;,\gamma_{4}^{u}\;\right)\right]\;<\gamma_{t}\right),$$
where $\gamma_{t}$ = $2^{2R/W}-1$ is the threshold SNR, *R* is the threshold data rate, *W* is the channel bandwidth, for OMA-based wireless network, the channel bandwidth for each node is, $W_{k}=\frac{W}{N}$ , *N* is the number of nodes [32]. Let $\gamma_{a}\;=\;min\left(\gamma_{1}^{u},\gamma_{2}^{u}\right)$ and $\gamma_{b}\;=\;min\left(\gamma_{3}^{u},\gamma_{4}^{u}\right)$, then (28) can be rewritten as
(29)$$Pr_{out}^{OMAC}\left(R\right)=Pr\left(min\left(\gamma_{a}\;,\gamma_{b}\;\right)<\gamma_{t}\right),$$

Using the probability theory of min(a,b), it can be further simplified as
(30)$$Pr_{out}^{OMAC}\left(R\right)=\underset{B1}{\underset{︸}{Pr(\gamma_{a}\;<\gamma_{t})}}\;+\;\left(1-Pr\left(\gamma_{a}\;<\gamma_{t}\right)\right)\underset{B2}{\underset{︸}{Pr(\gamma_{b}\;<\gamma_{t})}},$$

First term in (30) can be computed by substituting $\gamma_{a}\;=\;min\left(\gamma_{1}^{u},\gamma_{2}^{u}\right)$ as follows,
(31)$$B_{1}\;=\;Pr\left(x<\frac{\gamma_{t}}{\rho }\right)\;+\;\left(1-Pr\left(x<\frac{\gamma_{t}}{\rho }\right)\right)Pr\left(y<\frac{\gamma_{t}}{\rho }\right),$$
where $\rho \;=\;P_{t}/\sigma^{2}$ is the SNR of each node, $x={\left|h_{1}\right|}^{2}\;$ and $y={\left|h_{2}\right|}^{2}\;$. By Applying probability density function (PDF) for the variables *x* and *y* and then integrating, $B_{1}$ can be computed as
(32)$$B_{1}=1-\exp\frac{\gamma_{t}}{\rho }\left(\frac{\eta_{1}+\eta_{2}}{\eta_{1}\eta_{2}}\right),$$

Similarly, the second term can be determined as
(33)$$B_{2}=1-\exp\frac{\gamma_{t}}{\rho }\left(\frac{\eta_{3}+\eta_{4}}{\eta_{3}\eta_{4}}\right),$$

By substituting $B_{1}$ and $B_{2}$ in (30), the outage probability of an uplink can be determined as
(34)$$Pr_{out}^{UL}\left(R\right)=1-\exp\frac{\gamma_{t}}{\rho }\left(\eta_{a}+\eta_{b}\right),$$
where $\eta_{a}$ = $\left(\frac{\eta_{1}+\eta_{2}}{\eta_{1}\eta_{2}}\right)$ and $\eta_{b}$ = $\left(\frac{\eta_{3}+\eta_{4}}{\eta_{3}\eta_{4}}\right)$.

### 4.2. Downlink Outage Probability: A NOBC Link

In the downlink, the near node first decodes the far node signal, and it will be removed then the near node decodes its own signal.

Using (29) and (12), the end-to-end outage probability of the far node pair can be written as
(35)$$Pr_{out}^{E-to-E}\left(R\right)=Pr\left(min\left(min\left(\gamma_{a},\gamma_{b}\right),\gamma_{m}^{d}\right)<\gamma_{q}\right);\;m\in \left(3,4\right),$$
where $\gamma_{q}$ = $2^{3R}-1$, Let $\gamma_{c}\;=\;min\left(\gamma_{a},\gamma_{b}\right)$, then it can be further modified as
(36)$$Pr_{out}^{E-to-E}\left(R\right)=Pr\left(\gamma_{c}\;<\gamma_{q}\right)\;+\;\left(1-Pr\left(\gamma_{c}\;<\gamma_{q}\right)\right)Pr\left(\gamma_{m}^{d}\;<\gamma_{q}\right),$$
where m∈(3,4). After simplifying, the end-to-end outage probability of the proposed wireless network with ABS can be determined as
(37)$$\begin{array}{c} Pr_{out}^{E-to-E}\left(R\right)=1-\exp\frac{\gamma_{q}}{\rho }\left(\eta_{a}+\eta_{b}\right)\times \\ \left(e^{\frac{-\gamma_{q}}{\left(\rho_{2}-\rho_{1}\gamma_{q}\right)}}+\frac{e^{\frac{-\gamma_{q}}{\left(\rho_{2}-\rho_{1}\gamma_{q}\right)}}}{\left(1-\beta \right)}\left(1-e^{\frac{-\gamma_{q}}{\left(\rho_{2}-\rho_{1}\gamma_{q}\right)}\left(1-\beta \right)}\right)\right), \end{array}$$

**Proof.** See Appendix B. □

## 5. Bit Error Rate Analysis

In this section, average BER performance of the proposed wireless networks is analyzed. As in the previous section, error rate performance is analyzed in two phases such as the Uplink–OMAC and the downlink–NOBC. Finally, the end-to-end BER of the proposed network is analyzed.

### 5.1. Uplink—OMAC

Since the proposed wireless network operates with PLNC protocol, pair of nodes can transmit the data in a single time slot. Thus, the near node pair and far node pair send the data to the relay in the first and second time slots, respectively. Then, the relay node performs the bitwise PLNC operation $b_{r_{1}}=b_{1}⊕b_{2}$ where $b_{r_{1}}$ is the PLNC bit generated by XORing $b_{1}$ (first near node bit) and $b_{2}$ (second near node bit) at the relay. In other hand, it can be produced by multiplying the near node transmitted symbols as $x_{r_{1}}=x_{1}\times x_{2}$. Using ML detection rule, the symbol can be detected at the relay as given as [30].
(38)$$\begin{array}{c} \mathrm{if}\;\exp\left(\frac{-X\left(1,1\right)}{\sigma^{2}}\right)+\exp\left(\frac{-X\left(-1,-1\right)}{\sigma^{2}}\right)>\\ \exp\left(\frac{-X\left(1,-1\right)}{\sigma^{2}}\right)+\exp\left(\frac{-X\left(-1,1\right)}{\sigma^{2}}\right)\\ {\hat{x}}_{r_{1}}=1\;\mathrm{else}\;{\hat{x}}_{r_{1}}=-1, \end{array}$$
where $X\left(x_{1},x_{2}\right)={\left|z_{r,1}\;-\;\sqrt{P_{t}}x_{1}h_{1}\;+\;\sqrt{P_{t}}x_{2}h_{2}\right|}^{2}$. By taking logarithm of (38) and simplifying, it can be written as
(39)$$\begin{array}{c} \min\left(X(1,1),X(-1,-1)\right)>\min\left(X(1,-1),X(-1,1)\right)\\ {\hat{x}}_{r_{1}}=1\;\mathrm{else}\;{\hat{x}}_{r_{1}}=-1, \end{array}$$

Similarly, the relay decodes the second time slot received signal, $z_{r,2}$. Using NOMA scheme, these two signals are combined with different power levels, then the resulting signal forwarded to all four nodes in the third time slot simultaneously. The upper bounded instantaneous symbol error probability (SEP) of the OMAC link in the first time slot can be written as [30]
(40)$$p_{OMAC_{1}}<Q\left(\sqrt{2\gamma_{1}^{u}}\right)+Q\left(\sqrt{2\gamma_{2}^{u}}\right),$$
where Q(.) is the Q-function. Please note that the SEP and BER are same for the BPSK modulation. Similarly, in the second time slot it can be written as
(41)$$p_{OMAC_{2}}<Q\left(\sqrt{2\gamma_{3}^{u}}\right)+Q\left(\sqrt{2\gamma_{4}^{u}}\right),$$

By combining (40) and (41), the average SEP of the OMAC link can be computed as
(42)$$E\left[p_{OMAC}\right]<E\left[p_{OMAC_{1}}\right]+E\left[p_{OMAC_{2}}\right]$$
where E(.) is the expectation operator. By substituting the respective PDFs and then integrating, the simplified SEP can be written as
(43)$$E\left[p_{OMAC}\right]<\psi_{1}\left({\bar{\gamma }}_{1}^{u}\right)+\psi_{2}\left({\bar{\gamma }}_{2}^{u}\right)+\psi_{3}\left({\bar{\gamma }}_{3}^{u}\right)+\psi_{4}\left({\bar{\gamma }}_{4}^{u}\right)$$
where $\psi_{i}\left({\bar{\gamma }}_{i}^{u}\right)=\left(1-\sqrt{{\bar{\gamma }}_{i}^{u}/\left({\bar{\gamma }}_{i}^{u}+1\right)}\right)/2,i\in \left(1,2,3,4\right)$ is the average BER, ${\bar{\gamma }}_{i}^{u}=P_{t}{\tilde{h}}_{i}/\sigma^{2}$ is the average SNR and ${\tilde{h}}_{i}=E(|h_{i}{|}^{2})$ is the average channel gain.

### 5.2. Downlink—NOBC

At destination, near node pair apply SIC operation, so it first decodes far node pair signal by applying zero forcing equalization.
(44)$${\tilde{x}}_{r_{2}}\;=\;\frac{z_{n_{i}}}{g_{i}};\;i\in 1,2),$$
(45)$${\tilde{x}}_{r_{2}}\;\approx \;\alpha_{2}\sqrt{P_{r}}{\hat{x}}_{r_{2}}+in_{i};\;i\in 1,2),$$
where $in_{i}=\left(\alpha_{1}\sqrt{P_{r}}{\hat{x}}_{r_{1}}\;+\;\frac{n_{n_{i}}}{g_{i}}\right)$ is the interference plus noise. Far node 1 estimate ${\tilde{x}}_{4}$ by multiplying the detected symbol ${\tilde{x}}_{r_{2}}$ with its own message $x_{3}$. Similarly, far node 2 estimate ${\tilde{x}}_{3}$. After removing far node pair signal, near node pair decode its own signal.
(46)$${\tilde{z}}_{n_{i}}\;=\;\left({\hat{x}}_{r_{3}}-{\tilde{x}}_{r_{2}}\right)g_{i}+n_{n_{i}}\;i\in \left(1,2\right),$$
(47)$${\tilde{x}}_{r_{1}}\;\approx \;\alpha_{1}\sqrt{P_{r}}{\hat{x}}_{r_{1}}+\frac{n_{n_{i}}}{g_{i}};\;i\in 1,2),$$

The instantaneous end-to-end BER of the proposed wireless network can be calculated for each node.
(48)$$p_{E-to-E_{i}}\;=\;p_{MAC}\left(1-p_{NOBC_{i}}\right)+p_{NOBC_{i}}\left(1-p_{MAC}\right);\;i\in \left(1,2,3,4\right),$$
where $p_{NOBC_{i}}=Q\left(\sqrt{2\gamma_{i}^{d}}\right),\;i\in \left(1,2,3,4\right)$ is the instantaneous BER of each node and the average BER of each node is given as
(49)$$\chi_{i}\left({\bar{\gamma }}_{i}^{d}\right)=\left(1-\sqrt{{\bar{\gamma }}_{i}^{d}/\left({\bar{\gamma }}_{i}^{d}+1\right)}\right)/2,\;i\in \left(1,2,3,4\right),$$

Then, the overall BER of the proposed wireless network can be computed as given by
(50)$$p_{E-to-E}=\sum_{n=1}^{4}p_{E-to-E_{i}},$$

Using (40), (48) and (49), the average overall end-to-end BER of the proposed wireless network can be computed as
(51)$$E\left[p_{E-to-E}\right]=2p_{A}+p_{B}-2\left(p_{C}+p_{D}+p_{E}+p_{F}\right),$$
where $p_{A}=\psi_{1}\left({\bar{\gamma }}_{1}^{u}\right)+\psi_{2}\left({\bar{\gamma }}_{2}^{u}\right)+\psi_{3}\left({\bar{\gamma }}_{3}^{u}\right)+\psi_{4}\left({\bar{\gamma }}_{4}^{u}\right)$, $p_{B}=\chi_{1}\left({\bar{\gamma }}_{1}^{d}\right)+\chi_{2}\left({\bar{\gamma }}_{2}^{d}\right)+\chi_{3}\left({\bar{\gamma }}_{3}^{d}\right)+\chi_{4}\left({\bar{\gamma }}_{4}^{d}\right)$,$p_{C}=\left(\psi_{1}\left({\bar{\gamma }}_{1}^{u}\right)\chi_{1}\left({\bar{\gamma }}_{1}^{d}\right)\right)+\left(\psi_{1}\left({\bar{\gamma }}_{1}^{u}\right)\chi_{2}\left({\bar{\gamma }}_{2}^{d}\right)\right)$, $p_{D}=\left(\psi_{2}\left({\bar{\gamma }}_{2}^{u}\right)\chi_{1}\left({\bar{\gamma }}_{1}^{d}\right)\right)+\left(\psi_{2}\left({\bar{\gamma }}_{2}^{u}\right)\chi_{2}\left({\bar{\gamma }}_{2}^{d}\right)\right)$,$p_{E}=\left(\psi_{3}\left({\bar{\gamma }}_{3}^{u}\right)\chi_{3}\left({\bar{\gamma }}_{3}^{d}\right)\right)+\left(\psi_{3}\left({\bar{\gamma }}_{3}^{u}\right)\chi_{4}\left({\bar{\gamma }}_{4}^{d}\right)\right)$, $p_{F}=\left(\psi_{4}\left({\bar{\gamma }}_{4}^{u}\right)\chi_{3}\left({\bar{\gamma }}_{3}^{d}\right)\right)+\left(\psi_{4}\left({\bar{\gamma }}_{4}^{u}\right)\chi_{4}\left({\bar{\gamma }}_{4}^{d}\right)\right)$.

## 6. Numerical Results

In this section, sum capacity maximization of the downlink NOMA with ambient backscattering is presented. Furthermore, to enhance the throughput and reliability of each wireless link, outage and BER performance of the proposed wireless network is analyzed in uplink, downlink and then end-to-end level. Simulation parameters are given in Table 2.

Sum capacity of the downlink NOMA with ABS is shown in Figure 3a. Sum-rate of the proposed wireless network is maximized through the optimal power allocation. Simulation parameters are reflection coefficient β=0.8, stopping criterion’s $\epsilon_{1}$ = given SNR and $\epsilon_{2}$ = 0.01. It is assumed that the interference from ABS is zero. From this figure, it is clear that the sum capacity of the downlink NOMA with ABS is better than the downlink OMA. At 4 dB SNR, the sum capacity of the proposed network is 4 b/s/Hz whereas the wireless network with OMA achieves 2 b/s/Hz. Therefore, data rate of the proposed network is highly improved when compared to OMA network.

Sum capacity of the proposed wireless network is shown in Figure 3b for various reflection coefficient. It is observed that sum capacity of the proposed network is improved when the reflection coefficient is increases. At 6 dB SNR, sum capacity of the proposed network is 3 b/s/Hz at reflection coefficient of 0.2 whereas 4 b/s/Hz is achieved at reflection coefficient of 0.5.

The outage probability of the proposed wireless network with ABS is shown in Figure 4a. The channel variances are chosen as $\eta_{n_{1}}=d_{1}^{-\alpha }$ and $\eta_{f_{1}}=d_{3}^{-\alpha }$ [18]. Without loss of generality, 2/3 times of the transmit power is allocated to the far node and the rest of the power is allocated to the near node. The threshold data rate R=1 is considered. It is observed that the outage of the far node is highly reduced when the proposed network is operating with ABS. At target outage of $10^{-2}$, the downlink NOMA with ABS is require 13 dB SNR, whereas the downlink NOMA without ABS need around 25 dB SNR, thus 12 dB gain is achieved in proposed network using ABS.

Outage probability of the proposed network is compared with OMA network in Figure 4b. Data rate is varied from 0.5 to 1. At target outage of $10^{-2}$, proposed network need 30 dB SNR whereas network with OMA need 35 dB SNR. It is noted that when increase the data rate, outage of the wireless network is increased.

The outage performance of the proposed wireless network in uplink OMAC level is shown in Figure 5a. Similarly, the outage performance of the proposed wireless network in downlink NOMA level is shown in Figure 5b. It is observed that the outage performance of the downlink network is better than the uplink network this is because of the better downlink temporal efficiency. At target outage of $10^{-2}$, proposed uplink network need 25 dB SNR whereas downlink network need 22 dB SNR. Thus, downlink has 3 dB SNR gain than the uplink network.

Figure 6a shows the average BER performance of the proposed wireless network. BPSK modulation is considered in this simulation. Power allocation for the near node and the far node is splitted as 2/3 and 1/3 times of the total transmit power. Simulation results are used to validate the analytical results. At perfect SIC, the proposed wireless network ensures the reliability of each node. Further it is noted that only 2 dB additional SNR needed for the far node to achieve the near node reliability. For the target BER of $10^{-2}$, the uplink OMA need 17 dB SNR, end-to-end near node need 22 dB, far node need 24 dB SNR and overall network need 27 dB SNR.

In this proposed network, far node pair act as an IoT reader. It is achieved after far node removing its own information. With support of ABS, far node SNR gain in improved. Thus, far node pair used either as a IoT reader or mobile user. Average BER of the proposed network is shown in Figure 6b in the presence of interference. It is noted that both near nodes and far nodes operating with similar performance. Furthermore, it is observed that the effect of interference is dominant at high SNR regime.

## 7. Conclusions

In this paper, sum capacity maximization of the proposed wireless network is analyzed in the presence of ambient backscattering. The sum capacity of the NOMA scheme is compared with the conventional OMA scheme. Moreover, the end-to-end outage performance of a proposed wireless network is analyzed at different data rates. Simulation results conclude that proposed wireless network achieves better throughput performance in the presence of backscatter devices. Furthermore, the average BER performance of the proposed network ensure the reliability of each of the user link.

## Figures and Tables

**Figure 1 sensors-21-07589-f001:**
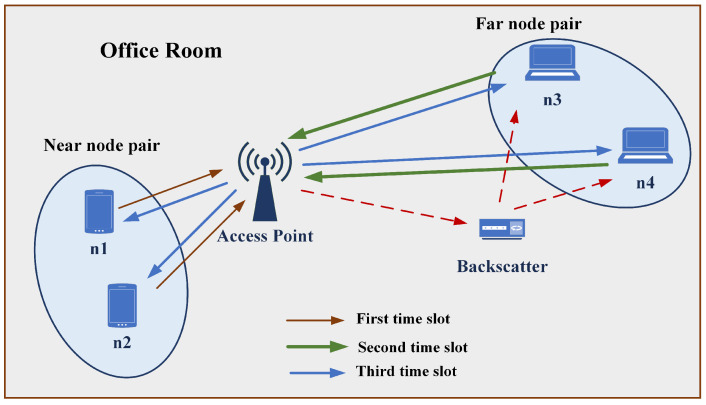
A two pairs NOMA-PLNC-based wireless network with ambient backscattering.

**Figure 2 sensors-21-07589-f002:**
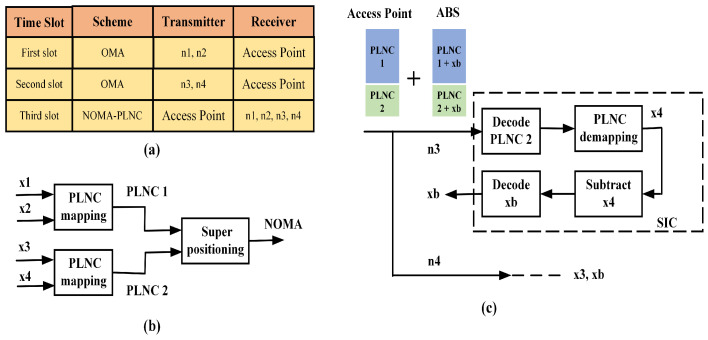
(**a**) Time slot management (**b**) Relay signal detection (**c**) SIC operation at far node.

**Figure 3 sensors-21-07589-f003:**
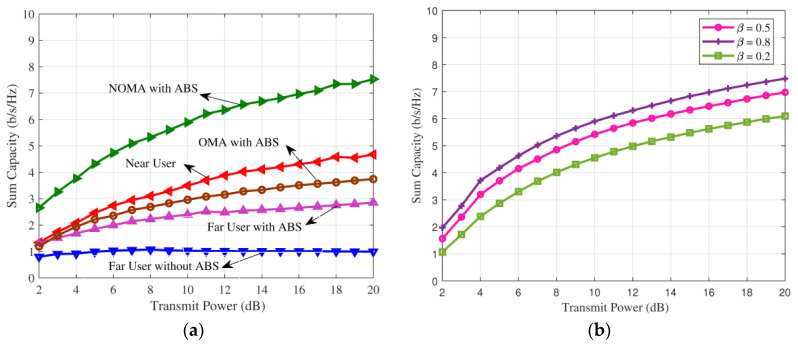
(**a**) Sum capacity of the downlink NOMA with ABS is compared with downlink OMA. (**b**) Sum capacity of the downlink NOMA with various reflection coefficients.

**Figure 4 sensors-21-07589-f004:**
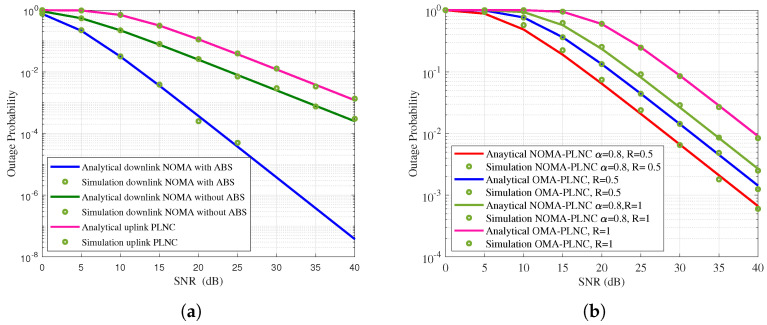
(**a**) Outage probabilityof the proposed network with ABS. (**b**) Outage probability of the proposed network compared with OMA.

**Figure 5 sensors-21-07589-f005:**
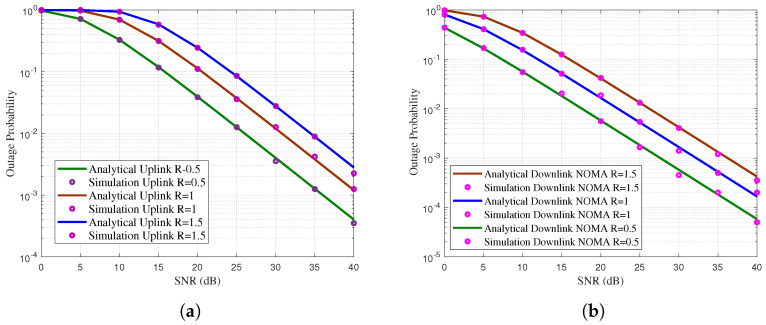
(**a**) Outage probability of the proposed uplink network. (**b**) Outage probability of the proposed downlink network.

**Figure 6 sensors-21-07589-f006:**
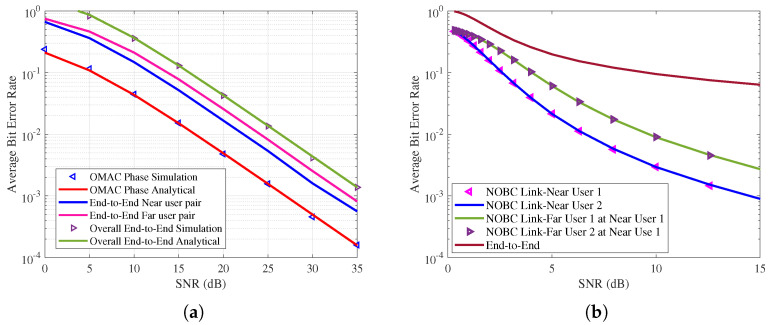
(**a**) Average BER performance of the proposed wireless network with perfect SIC. (**b**) Average BER performance of the proposed wireless network in the presence of the interference.

**Table 1 sensors-21-07589-t001:** Comparison of NOMA and NOMA with PLNC.

S.No.	Parameters	Uplink OMA and Downlink NOMA	Uplink OMA and Downlink NOMA with PLNC
1.	No of nodes	4	4
2.	Time slots	5	3
3.	SIC operation	3	1
4.	Complexity	high	low
5.	nodes connectivity	2 for 3 time slots	4 for 3 time slots

**Table 2 sensors-21-07589-t002:** Simulation Parameters.

S.No.	Parameters	Symbols	Value
1.	No of nodes	*N*	4
2.	Distance between the near node 1 and the AP	$d_{1}$	2 m
3.	Distance between the far node 1 and the AP	$d_{3}$	10 m
4.	Path loss exponent	α	2
5.	Reflection coefficient	β	0.8
6.	Threshold data rates	*R*	0.5, 1, 1.5

## Data Availability

Study did not report any data.

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
