# Peer review of "Backscatter Assisted NOMA-PLNC Based Wireless Networks"

_sensors, 2021, doi:10.3390/s21227589_

Round 1
Reviewer 1 Report
The idea to use a passive backscatter system to help IoT wireless communication is good.
I’m not very clear how a passive relay work as the relay is just reflecting the RF signal instead of amplifying the signal.
It is better to include the basic idea (i.e., why a backscatter relay can help the network performance) in the abstract as well as introduction section.
Too many abbreviation in the title, which may confuse the reader.
The simulation setup is not very clear. It will be better to explain the parameters in table 2.
Looks to me, some important ambient backscatter reference are missing. Please check recent year backscatter paper in SIGCOMM and NSDI.
Author Response
Thank you very much for your positive feedback and insightful comments on our manuscript. We provide our detailed response in the attached PDF file.

Reviewer 2 Report
The paper considered performance analysis of Backscatter Assisted NOMA Based IoT Wireless Networks along with solid mathematical derivations. These are the comments to improve the quality of this paper as below.
- Introduction section need to be improved, please add few more recent references with closely related topics of NOMA, backscattering, for example.
[R1] "NOMA in Cooperative Underlay Cognitive Radio Networks Under Imperfect SIC," IEEE Access, vol. 8, pp. 86180-86195, 2020
[R2] "Performance Evaluation of Relay-Aided CR-NOMA for Beyond 5G Communications," in IEEE Access, vol. 8, pp. 134838-134855, 2020
[R3] "Performance Analysis of Clustering Car-Following V2X System with Wireless Power Transfer and Massive Connections," in IEEE Internet of Things Journal, doi: 10.1109/JIOT.2021.3070744.
- In page 4, below eqn.4, check the notation for the power allocation coefficient, further, please check all the notations, symbols and abbreviations used in this paper carefully and then modify accordingly.
- Please check all figures and Table captions in this manuscript and modify as per format.
- In numerical sections, explanation for figures 3.a, 3.b, 5.a, 5.b is not clear, please rewrite the discussion for those figures in details. Further, write the separate para in numerical section about significance of this proposed work in the IoT wireless network.
- Few equations are not written in standard format, please check all the equations once again carefully and modify.
Author Response

(The authors gave the same response as above.)

Reviewer 3 Report
Dear Authors:
Here in the following You can find my comments for your paper.
The paper needs to be carefully reviewed for what concerns the english language.
I suggest to enlarge the descriptions (even with using additional figures) of the considered methods that brought the conclusions and/or the results presented in the text.
In the introduction, in line 61: I suggest to move the table after the end of this phrase.
In the section 2: This section needs to be carefully revised. In fact, the system model and the SIC diagram is not described properly. I suggest to follow an approach as in the reference [4], i.e., a presentation with a complete diagram where the environment is deeply described, also by using schematics similar to the one in Fig. 5 in [4].
The whole set of parameters used in the mathematical approach should be clearly explained in the text, as well as in the diagrams.
Fig. 2 does not clearly explain the concept of SIC. It would be more useful if a flow chart is deployed, with a description in the text.
In subsection 2.1: The time index (or the time variable) is not considered in the equations. I suggest You to declare Your choice in the text, declaring if the time domain should be considered discrete or continuous.
I noticed that the transmitted power P_t is taken equal for each node. This fact should be declared in the text.
The channel coefficients are taken to be narrowband with a exponential distribution. Could You add a description or a reference where this choice has been motivated?
Equation (3) contains symbols which are not defined/described in the text (for example \hat{x}_1 and so on). More over, it is not clear what is the effect of \times operator. In line 124 the signal x_{r3} mismatches with the notation in Eq. (4), where the \hat is used.
In section 3. The acronym SINR should be defined before its usage. Also, I suggest to use a different symbol to identify the SINR with respect to the SNR.
In section 4. The SNR threshold should be mathematically described in its terms. For example, R is not defined in the text, but it is used in the equations.
Equation (30) has two callouts labeled I and II, while in the text the quantities are referred to be B_1 and B_2. Could You correct the notation?
Just before Equation (32), is it possible to add in the text a kind of trace for the mathematical manipulations done to achieve the result (32)?
Thank You for the attention.
Author Response

(The authors gave the same response as above.)

Round 2
Reviewer 1 Report
The following backscatter papers are still missing:
Verification and Redesign of OFDM Backscatter [NSDI'21]
VMscatter: A Versatile MIMO Backscatter [NSDI'20]
Leveraging Ambient LTE Traffic for Ubiquitous Passive Communication [SIGCOMM'20]
Author Response
We would like thank to the reviewer for your kind support, which make this paper as worth full among research community. Please see the attachment for response of the second round review comments.

Reviewer 2 Report
It can be accepted in current form.
Author Response
Thank you very much for your positive feedback and accepting our revised manuscript in its current form.

Reviewer 3 Report
Dear Authors,
Firsly, I would like to thank You for your accurate reply to my comments. I'm glad for having given a little contribution to Your manuscrip.
Some typos and grammar errors are still present in the text, so I suggest you to send the manuscript to an english expert.
Also, I found that the \times (the cross used as operator) in equation 3 has not a description in the text.
Best regards.
Author Response
We would like thanks to the reviewer for appreciating our efforts. We have addressed second round comments in the revised manuscript. Please see the attachment for response of second round review comments.
